# Psychometric Characteristics of the Physical Activity Enjoyment Scale in the Context of Physical Activity in Nature

**DOI:** 10.3390/ijerph16244880

**Published:** 2019-12-04

**Authors:** Julio Fuentesal-García, Antonio Baena-Extremera, Jesús Sáez-Padilla

**Affiliations:** 1Pontificia University of Comillas, Calle de Alberto Aguilera, 23, 28015 Madrid, Spain; 2Faculty of Education Sciences, University of Granada, Avenida Fuente Nueva s/n, 18071 Granada, Spain; abaenaextrem@ugr.es; 3Faculty of Education, Psychology and Sport Sciences, University of Huelva, Avenida Tres de Marzo s/n, 21071 Huelva, Spain

**Keywords:** enjoyment, physical activity in nature, outdoor education, teenagers

## Abstract

The aim of this study was to analyze the psychometric properties of the Physical Activity Enjoyment Scale applied to different contexts for initial or original use, such as in the context of physical activity in nature. In order to do this, we carried out a study at some primary and secondary schools located in western Andalucía (Spain), with students aged 9–12 years old (M = 11.22; SD = 1.07). Therefore, a sample of 206 students in Study 1 (98 boys = 47.8%; 107 girls = 52.2%) and 455 students in Study 2 (228 boys = 50.1% 227 girls = 49.9%) was used. The students of the two groups that belong to the study created a program related to Physical Activity in Nature. Descriptive, exploratory, and confirmatory analyses were conducted. We also analyzed several other factors, such as internal consistency, composite reliability, average extracted variance, and convergent validity. Afterward, differences according to gender and school year were also studied. The data showed the need to eliminate many of the items from the original scale, giving, as a result, a model of six items that satisfactorily fit into the confirmatory analysis for the use of physical activity in nature. The ANOVA statistical test, used to analyze sex and school year, did not show any tangible differences between the target groups. Thanks to its application, we note that the PACES instrument cannot be applied as-is; some items must be removed or modified. Therefore, we must obtain a new, more specific instrument for these types of incipient practices undertaken in natural environments.

## 1. Introduction

We have, in recent years, witnessed an important increase in physical activity levels among the general population [1]. The rise in the number of hours devoted to physical activity, plus the variety of spaces and places where such activities can be carried out, have transformed the outdoors and nature into one of the main environments for this type of activity [2]. Enjoyment and pleasure are some of the reasons behind increases in the practice of physical activity. Enjoyment in the context of physical activity is specifically defined as a positive cognitive or physiologic state that involves feelings of pleasure and fun related to the practice of a physical activity [3]. The outdoors has become an ideal place to escape from society, routine, and life stress, and are thus an ideal place to return to the essence of being oneself. Being in contact with nature also brings with it multiple personal benefits [4]. Twenty-first-century sports practice is also particularly focused on enhancing the experience of strong emotions and exciting sensations, based on the idea that intense enjoyment drives athletes to test their personal limits (challenges) by exploring strange and unusual spaces—in short, searching for adventure and risk [5]. This suggests that people are spending more free time engaging in physical activities in a natural environment; evidence shows that enjoying these natural environments increases our health benefits [6].

Physical activity in nature entails sports–physical activities embodied in educational legislation, which have begun to be practiced in schools with the aim of instilling future habits in the population [4]. One of the main characteristics of these outdoor sport disciplines is enjoyment, which is also one of the central motivations for people to practice sports [7]. Following Scanlan and Symons [8], enjoyment involves feeling pleasure, joy, and fun. In this regard, Granero-Gallegos et al. [9] have shown the relationship, for instance, between students “enjoying sports practice and subjects” through intrinsic motivation (according to self-determination theory [10]), the importance that students attribute to physical education, and, more importantly, the intention of doing physical activity in the future and, therefore, the intention to do physical activity in nature. Enjoyment has been studied from different theoretical perspectives in the search of an explanation for its relationship with physical and sports practice, as enjoyment is perceived to be a key factor in this regard [11]. Therefore, enjoyment has been associated, among others, with perceived athletic competition or with the performance of future physical activity.

A series of instruments have been used to assess enjoyment in physical activity (e.g., the Subjective Exercise Experiences Scale (SEES) [12] and the Enjoyment Scale as a dimension of achievement motivation [13]). In this study, we used the instrument proposed by Molt et al. [14], since it is one of the most relevant models in terms of psychometric results. This instrument includes 16 items and has been adapted for adolescents; a series of studies reveal an interesting relationship between enjoyment and the different versions of intrinsic motivation [15]. Bearing in mind the increase in the number of people doing physical activity in nature, for its competitive, recreational, health, and leisure aspects, we propose the use of Physical Activity Enjoyment Scale (PACES) for school-age youth and adolescents, in the context of physical activity in nature. According to the existing literature, the PACES is an instrument designed for use with physical and sporting activities in conventional sports facilities [16]. Based on our investigations, we advanced the hypothesis that the PACES would perfectly adapt to physical activity practiced in the context of nature and open air and obtained satisfactory results when analyzing the psychometric characteristics of a sample of students aged 9–13 years old [17]. Thus, the object of this study is to analyze the psychometric characteristics of the PACES in a sample of 9–13 years old students in the context of outdoor physical activity practice using confirmatory procedures, along with a gender and school year analysis. It is worth pointing out that these practices use formative and educational approaches rather than sport-related ones [5,18]. Moreover, there is less research on our study variable in the field of physical activity in nature and outside educational contexts.

## 2. Materials and Methods

### 2.1. Study 1

#### 2.1.1. Design

In terms of the sample, a non-probabilistic convenience design was applied, based on the subjects that could be accessed. The design used was non-experimental, cross-sectional, and descriptive.

#### 2.1.2. Participants and Procedure

A total of 205 students (98 boys and 107 girls) participated in this study, with an age range of 12–13 years old (M age = 12.22, SD = 0.57), all being 1st and 2nd-year high school students in different educational centers in western Andalusia, Spain. Students, parents, and school management were briefed on the data collection. Once authorizations were obtained, instructions were given, and doubts were dispelled prior to filling in the questionnaires. The questionnaires were answered voluntarily and anonymously upon arrival at the camp site (Albergue Campamento de Andévalo Aventura SLL) in approximately 20 min. The Andévalo Camp is a company with a specialization in Adventure Sport and Physical Activity in the Natural Environment. Further, all students usually go to this camp each year for a period of two to seven days. Consequently, the students know this camp and these activities. Thus, the students should be able to answer this questionnaire from the point of view of physical activity in the natural environment and not from the point of view of sports activities. The researcher insisted on this when it came to responses.

#### 2.1.3. Instrument and Application to the Physical Activity in Nature Context

We used the PACES [15] adapted to a Spanish context by Moreno et al. [16]. This scale measures enjoyment during physical activity through 16 items, which are preceded by the phrase “When I am active…” This instrument measures enjoyment and bipolar enjoyment, through statements such as “I enjoy it”, “I’m bored”, “It’s very exciting”, and “I don’t like it”. The answers were collected on a Likert scale from 1 (totally disagree) to 5 (totally agree).

This application was analyzed by a group of four experts (Lynn, 1986) in education and physical activity in nature in order to ensure the appropriate design of the items for what the construct was intended to measure; in this way, the original meaning was maintained [19]. These experts were given a table with the item specifications [19], which collected the semantic definition of the construct to be assessed and that of its component. Next, they were shown the list of items after they were adapted so that they could give their opinion on the items’ suitability and understandability on a scale from 1 (totally disagree) to 4 (totally agree). Furthermore, they were provided a section to write general notes and observations on each of the items and compose an alternative version for each item if they saw fit. The items with <2.5 mean scores, both in terms of suitability and understandability, were revised. If an item was not classified by at least three of the four experts within the theoretical dimensions of the scale, it was revised again to analyze potential problems before proposing alternative wording that would cover the theoretical dimension in a more clear and accurate way. Bearing in mind that this scale was created by the author, this scale has two dimensions of seven items each, which is the theoretical dimension to which each item belongs. The overall agreement of the four experts on the suitability and understandability of the items was measured using the intraclass correlation coefficient (ICC) based on a mixed effects model and assuming an absolute agreement definition; the values obtained were ICC = 0.77 in suitability and ICC = 0.80 in understandability.

Furthermore, the interquartile range was the standard used to measure dispersion among the four experts’ agreements. If the difference between percentile 3 and percentile 1 equaled 0 or 1, the item was accepted and/or slightly modified; if this difference was between 1 and 2, the item was revised and reformulated, and if it was higher than 2, the dispersion between experts was too high, and the item was rejected. Finally, the experts’ comments on the instructions and their wording resulted in minor changes. Once all changes requested by the experts had been made, the final version was administered to sixty-five 12–13-year-old high school students. They confirmed their full understanding of the items, and, after a final revision by the research team, we obtained the final version of the PACES adapted to physical activity in nature. This instrument will be only applicable to activities in the natural environment, carried out in that context. Thus, people who are going to respond to this questionnaire should be immersed in nature.

#### 2.1.4. Data Analysis

The psychometric properties of the Physical Activity Enjoyment Scale adapted to physical activity in nature for school students were analyzed through the statistical analysis of the items (i.e., an exploratory factor analysis) SPSS Statistics 21.0 software (IBM, University of Chicago, USA) was used for data analysis.

### 2.2. Study 2

#### 2.2.1. Design

In terms of the sample, a non-probabilistic convenience design was used, based on the subjects that could be accessed. The design we used was non-experimental, cross-sectional, descriptive, and exploratory.

#### 2.2.2. Participants and Procedure

A total of 455 students (228 boys and 227 girls) participated in this study, the age range being 9–13 years old (M_age_ = 11.22, SD = 1.07); all were primary school students (5th and 6th year) and 1st and 2nd year high school students in different educational centers in western Andalusia, Spain. Students, parents, and school managements were briefed on data collection. Once authorizations were obtained, instructions were given, and doubts were dispelled prior to filling in the questionnaires. The questionnaires were answered voluntarily and anonymously upon arrival at the camp site (Albergue Campamento de Andévalo Aventura SLL) in approximately 20 min.

#### 2.2.3. Instrument

We used the Physical Activity Enjoyment Scale (PACES) adapted to physical activity in nature described in Study 1.

#### 2.2.4. Data Analysis

We carried out a confirmatory factor analysis (CFA), and reliability was measured through Cronbach’s alpha. Average variance was extracted, and composite reliability and McDonald’s ω were also estimated. Convergence validity and gender invariance, as well as gender and school year differences, were measured with Student’s t and ANOVA tests. Data analysis was carried out with SPSS Statistics 21.0 and LISREL 8.80 [20].

## 3. Results

### 3.1. Study 1

#### 3.1.1. Descriptive Analysis

First, we carried out an analysis of each of the items in the scale following the suggestions of Carretero-Dios et al. [21]. In line with the contributions of Nunnally et al. [22], we analyzed whether the internal consistency of the scale increased with the elimination of any of the items, and the uniqueness necessary to keep an item inside a factor was studied. The corrected item-total correlation (C-ITC), coefficient, standard deviation (SD), and all answer options were used at some point. Moreover, skewness and kurtosis indices had to be close to 0 and <2 for these items to be accepted. Initially, we tried using only one factor, as done in the original versions of the study, but the data were not satisfactory (the Cronbach’s alpha value even less so). Consequently, we opted to distinguish between two factors: a negative valency and a positive valency of enjoyment (Table 1).

An analysis of the items and factors revealed that the alpha values are acceptable. Despite this, the SD results obtained for many of the items have problems (Items 1, 2, 3, 4, 7, 8, 9, 12, 13, and 15), as their values range between 0.770 and 0.957. Furthermore, the skewness values of the negative enjoyment items are above 2.21, and their kurtosis values are above 4.11. For positive enjoyment, many of the items (1, 4, 8, 15) show skewness (−2.38 to −2.92) and kurtosis values (3.24 to 9.39), which suggests that they should be eliminated. However, it is worth noting that the alpha value of each factor did not increase if the problematic items were deleted. Furthermore, the C-ITC of all the items showed values ≥0.32, so we assumed the possibility of using these items in the scale.

#### 3.1.2. Exploratory Factor Analysis (EFA)

An EFA for one factor was carried out, and the data ruled out this structure. Therefore, a two-factor EFA solution was carried out using principal component analysis (PCA), requiring a 0.40 minimum correlation for each item important within a factor [23]. The Kaiser–Meyer–Olkin measure was adequate (0.89), and Bartlett’s test was statistically significant (χ^2^ = 1658.55, *p* < 0.000), all of which verified the suitability of the EFA. The results confirm the two-factor extraction (Table 2). However, the explained variance was low, with a 39.2% value for the total scale.

### 3.2. Study 2

#### 3.2.1. Confirmatory Factor Analysis

Structural equation modeling was applied to study the psychometric properties of the PACES adapted to its original physical activity in nature dimension. A series of absolute and relative fit indices were estimated to assess the models [24,25]. For the absolute fit indices, we used the *p* value associated with the Chi-square statistic (χ^2^), the ratio between χ^2^ and degrees of freedom (d.f.) (χ^2^/d.f.), and GFI (goodness-of-fit index). For the relative indices, we analyzed the NNFI (non-normed fit index) and CFI (comparative fit index). The RMSEA (root mean square error of approximation) was also estimated as the incremental index. The parameters are considered significant when the associated t value is above 1.96 (*p* < 0.05).

First, a multivariate normal distribution analysis was carried out for this scale using a normality test based on the relative multivariate kurtosis (RMK) of PRELIS, LISREL 8.80. The PACES normalized multivariate kurtosis was 32.4 (Mardia–Based–Kappa = 0.726). The critical test value was 1.96 (5%). The test results rejected multivariate normality, which implies the use of robust estimators. In light of this, we used the weighted least squares (WLS) method in LISREL 8.80 [20]. The polychoric correlation matrix and the asymptotic covariance matrix were used as input for data analysis. A two-factor measurement model was hypothesized. The calculations revealed that the RMSEA values, as well as some of the factor loads (item 14 = 0.37) and individual reliability (R^2^ > 0.50) of many of the items were not suitable (Table 3). Following Byrne [26], the items with high values in standardized residuals (>±2.58) were considered for potential elimination.

These data, plus those in Table 1, support eliminating items with low values. The CFA values were χ^2^/d.f. = 4.75; *p* < 0.000; RMSEA = 0.09; ECVI = 1.223; NNFI = 0.950; CFI = 954; IFI; 0.954; GFI = 0.881. Thus, following Markland [25] and Levy and Hancock [27], we carried out a series of analyses of different models, as suggested by the data, and the items with a low factor load and low R^2^ were eliminated. The final result was a PACES-Outdoor Physical Activity (OPACT) with a two-factor model of six items whose EFA values were: χ^2^/d.f. = 0.65; *p* < 0.000; RMSEA = 0.005 (IC90% = 0.004, 0.006); ECVI = 0.057; NNFI = 1.002; CFI = 1.00; IFI = 1.001 and GFI = 0.998 (Table 4). 

Table 4 shows the fit indices of the two-factor six-item model, which was the only one with the minimum requirements to guarantee convergent validity [28] high standardized factor loads (>0.60), which are statistically significant (*t*-value > 1.96). Finally, in light of the low alpha values in the ordinal EFA of the scales in the correlation matrix, we also provide the EFA composite reliability and average variance extracted (AVE) values for each dimension (Table 5). The AVE reflects the total variance of the indicators collected by the latent construct; the higher the value, the more representative the indicators of the critical dimension to which they are loaded, considering the limitations of Cronbach’s alpha [29], especially when the variables include a low number of items [30] (like the case of the instrument analyzed in this study). McDonald’s ω was calculated to measure reliability since, unlike the alpha coefficient, McDonald’s ω takes into account the factor loads. Thus, the calculations are more stable and reflect the actual reliability level regardless of the number of items in the variable [29]. Internal consistency values (*α*) are considered suitable when they are in the 0.70–0.90 range [31].

In terms of convergent validity, the validity of indicators can be assessed based on the size of factor loads [32]. Thus, the NFI was 0.943 for the 16-item scale, whereas for the six-item instrument, this value was 0.998. The AGFI was 0.843 for the 16-item model and 0.991 for the five-item model. Moreover, as mentioned above, saturation was, in all cases, statistically significant (*t*-value > 1.96), which means that all indicators assess the same theoretical construct [33]. Finally, it is worth noting that all the items have high factor loads (R^2^ > 0.50).

#### 3.2.2. Gender and School Year Differences

Next, in order to analyze gendered differences, we carried out a Student’s *t*-test for the independent samples and an ANOVA to study school year differences. As seen in Table 6, no gender differences were found for the positive and negative enjoyment of the PACES in any of the independent variables, such as years.

#### 3.2.3. Invariance Analysis

A gendered invariance analysis was carried out (Table 7) to simultaneously test the equivalence of the factor structure of both sub-groups. No significant differences were found between Model 1 (model with no restrictions) and Model 2 (invariance in measurement loadings) (*p* = 0.536) or between Model 1 and Model 3 (invariant structural variances and covariances) (*p* = 0.378). However, statistically significant differences were observed between Model 1 and Model 4 (invariant measurement residuals) (*p* = 0.037). According to Byrne [26], the absence of statistically significant differences between Model 1 and Model 2 constitutes a minimum standard to accept the existence of invariance in the model, in this case, in terms of gender. Moreover, the decrease in the CFI values was also taken into account; they were <0.1 (ΔCFI contrast test) across the different models. Thus, following Cheung and Rensvold [34], the model was proven to be gender invariant.

## 4. Discussion

The main objective of this work was to analyze the psychometric characteristics of the PACES scale in the context of physical activity in nature for a group of primary and secondary physical education students. We have followed the process set out in Crocker et al. [35] related to the usefulness of this scale to measure enjoyment in different physical education areas; that is, we have used a bi-dimensional fit model for the construct.

Physical activity in nature is included in the physical education classes that the students in this study engage in during school hours [5]. These activities include orientation, hiking, outdoor games, gymkhanas, knotting, climbing, mountain biking, etc. [36]. Furthermore, other studies have proven the beneficial effects of these types of activities for students when physical activity is undertaken outdoors, as this change in environment improves aspects like satisfaction and fun in class, self-perception, and even social goals [37,38,39,40]. Consequently, it is essential to use the new instrument in this study since it will help promote the advancement of knowledge in the field of outdoor physical activity, physical education, and sports–physical activity.

The analyses carried out have shown that the original 16 item scale is not able to meet the objective of analyzing students in this context; thus, it is necessary to reduce the scale. First, the descriptive internal consistency and homogeneity values clearly demonstrate the need to modify the scale by eliminating some of the items that showed problems. This was then confirmed by the CFA, which clearly showed that the scale had to be reduced in order to fall within the fit indices’ acceptable values, and many of the items had factor loading and individual reliability problems. This reduction in the scale is also interesting for primary school students since, for them, it would be more suitable to assess variables through six rather than 16 questions if the results allowed for this, and their answers are likely to be more reliable in relation to longer questionnaires.

Another relevant aspect is that the scale did not show a good fit for one factor based on the original Molt et al. [15] version adapted to Spanish by Moreno et al. [16]. It should be noted that in other studies [41], the factorial structure of the scale was not analyzed, necessitating its future analysis in later works using broader samples.

However, another study [42] performed this analysis with the adolescent population, also finding problems in their one-dimensional adjustment. Their results [36] confirmed this one factor structure with the inactive adult population and obtained good values. In the present case, both the exploratory and, especially, the confirmatory analysis only allowed a two-factor version of enjoyment by distinguishing between positive and negative enjoyment.

For the ANOVA and invariance analyses, no gendered or school year differences were found. In terms of gender, enjoyment constitutes a particularly relevant factor in the study of behavior in physical activity, as enjoyment has been consistently shown to be related to female participation [28,43,44]. There are different studies that show that girls’ concern is to shoot boys; the scientific literature has previously shown that part of girls’ concerns when it comes to physical activity is not the enjoyment of physical activity but to perceive a better physical appearance along with controlling your weight. This leads them to not pay attention to enjoyment while doing physical activity, either in nature or outside it [45]. Along the same lines, another investigation [46,47], showed how student boys, unlike student girls, obtained better scores referring to levels of enjoyment when doing physical activity. Another reason is the critical moment of the transition from primary to secondary. However, in our research, there were no significant differences between boys and girls. Moreover, other authors [48,49] hold that enjoyment is a consistent predictor of physical activity among teenage girls. Recent studies confirm that girls have statistically lower levels of enjoyment and physical activity [18]. This lack of significant differences might be related to various factors.

Another factor is that a large part of the sample was made up of primary school students, and it is worth noting here that in this education stage, physical activity in nature is less frequent than in secondary school. This means that students have covered fewer subjects (physical activity in nature does not exist as such in primary school and is included only through games and sports); therefore, there is a very low level of gendered differences. Likewise, primary school teacher training courses are different from secondary education teacher training. Sports Science degrees include compulsory outdoor activity training [50,51], as opposed to primary school teacher degrees, where (in most cases) this subject is not taught. The training courses received by secondary school teachers address the contents of physical activities in the natural environment, unlike those of primary school teachers, for whom such courses are non-existent. This means that physical activity in natural contexts is virtually absent throughout primary school, except when teachers are particularly interested in this field, hence the difficulty in finding significant differences.

## 5. Conclusions

To conclude, it is worth noting that the PACES-OPACT (Outdoor Physical Activity) model has partially verified the hypothesis that it has not been fully adapted; thus, it has been necessary to create a six-item model. This model has a very good fit, as demonstrated by the CFA, and has high reliability and validity (see Table 5). The convergent validity values also provide these data with robustness.

This instrument advances the field of existing research in several ways:

(1)An instrument is applied in a different context than it was created for, which is the natural environment.(2)From this application, it is concluded that the PACES, in this context, cannot be applied as-is. Instead, some items must be eliminated and/or modified. Therefore, we obtained a new specific instrument for this type of practice.(3)An invariance analysis was carried out, something that had never before been done using any of these instruments. Therefore, new data are provided.(4)Finally, sex and course analysis were performed and have not been differentiated, something that gave different results in other works.

However, in terms of potential future perspectives, it would be good to contrast these results by differentiating a primary school sample and a secondary school sample from physical activity in nature taught exclusively at schools or exclusively outside schools [52]. It is important to note that the location of schools and the extra-mural outdoor activities that students may engage in, such as scouting groups, as well as closeness to natural environments, change the vision and knowledge of these elements [53]. Thus, we believe that although these data fully support this model’s application, it would be interesting to contrast these results with those of future studies.

## Figures and Tables

**Table 1 ijerph-16-04880-t001:** Descriptive, internal consistency and homogeneity statistics (*n* = 205)**.**

Scale:	M	SD	CCIT-c	α without ítem	Asymmetry	Kurtosis
*Positive Enjoyment (α = 0.786)*						
1. I enjoy it	4.67	0.770	0.51	0.76	−2.92	9.39
4. I find it pleasant	4.39	0.948	0.50	0.76	−1.75	2.87
6. It gives me energy	4.12	1.13	0.53	0.76	−1.31	0.983
8. It’s very exciting	4.46	0.916	0.57	0.75	−1.86	3.24
9. My body feels good	4.31	0.957	0.54	0.76	−1.50	1.85
10. I get something extra from it	3.87	1.19	0.45	0.77	−0.880	−0.030
11. It´s very exciting	3.97	1.23	0.45	0.77	−1.05	0.150
14. It produces strong feelings in me	3.23	1.35	0.32	0.80	−0.319	−0.960
15. I feel good	4.60	0.792	0.55	0.76	−2.38	5.97
*Negative Enjoyment (α = 0.700)*						
2. I´m bored	1.46	0.906	0.42	0.66	2.23	4.94
3. I don´t like it	1.35	0.857	0.39	0.67	2.75	7.36
5. It´s no fun at all	1.41	1.04	0.39	0.67	2.59	5.77
7. It depresses me	1.32	0.885	0.47	0.65	2.96	8.11
12. It frustrates me	1.35	0.881	0.46	0.65	2.76	7.22
13. It´s not at all interesting	1.31	0.839	0.35	0.68	2.91	7.98
16. I think I should be doing something else	1.49	1.00	0.39	0.67	2.21	4.11

**Table 2 ijerph-16-04880-t002:** Rotated component matrix (*n* = 205).

Scale:	*F1*	*F2*
*Positive Enjoyment (α = 0.786)*		
1. I enjoy it		0.411
4. I find it pleasant		0.518
6. It gives me energy		0.585
8. It´s very exciting		0.537
9. My body feels good		0.581
10. I get something extra from it		0.715
11. It´s very exciting		0.597
14. It produces strong feelings in me		0.554
15. I feel good		0.440
*Negative Enjoyment (α = 0.700)*		
2. I´m bored	0.549	
3. I don´t like it	0.531	
5. It´s no fun at all	0.505	
7. It depresses me	0.685	
12. It frustrates me	0.625	
13. It´s not at all interesting	0.479	
16. I think I should be doing something else	0.560	

**Table 3 ijerph-16-04880-t003:** Items individual reliability (*n* = 455).

Scale:	Load	R^2^
*Positive Enjoyment (α = 0.786)*		
1. I enjoy it	0.79	0.624
4. I find it pleasant	0.67	0.455
6. It gives me energy	0.65	0.423
8. It´s very exciting	0.77	0.600
9. My body feels good	0.66	0.429
10. I get something extra from it	0.49	0.238
11. It´s very exciting	0.55	0.307
14. It produces strong feelings in me	0.37	0.140
15. I feel good	0.80	0.632
*Negative Enjoyment (α = 0.700)*		
2. I´m bored	0.66	0.435
3. I don´t like it	0.71	0.500
5. It´s no fun at all	0.67	0.455
7. It depresses me	0.76	0.580
12. It frustrates me	0.70	0.486
13. It´s not at all interesting	0.65	0.423
16. I think I should be doing something else	0.62	0.386

**Table 4 ijerph-16-04880-t004:** Items’ individual reliability (*n* = 455).

Scale	Load	R^2^
Positive enjoyment (α = 0.712)		
1. I enjoy it	0.74	0.552
8. It’s very exciting	0.75	0.560
15. I feel good	0.84	0.702
Negative enjoyment (α = 0.716)		
3. I don’t like it	0.73	0.560
7. It depresses me	0.80	0.640
12. It frustrates me	0.75	0.566

**Table 5 ijerph-16-04880-t005:** Scale reliability and validity.

PACE—Five Item Model	Convergent Validity	AVE	Cronbach´s Alpha	McDonald’s ω
Positive Enjoyment	0.82	0.60	0.85	0.83
Negative Enjoyment	0.75	0.60	0.85	0.80

**Table 6 ijerph-16-04880-t006:** Variance analysis according to gender and school year.

PACES	Male (*n* = 228)		Female (*n* = 227)				5th Primary (*n* = 20)		6th Primary (*n* = 389)		1st ESO (*n* = 17)		2nd ESO (*n* = 29)			
	M	SD	M	SD	F	*p*	M	SD	M	SD	M	SD	M	SD	F	*p*
Positive Enjoyment	4.61	0.62	4.53	0.68	1.81	0.18	4.62	0.55	4.58	0.65	4.15	0.86	4.66	0.52	2.59	0.052
Negative Enjoyment	1.28	0.73	1.39	0.80	2.25	0.13	1.25	0.66	1.33	0.74	1.47	0.72	1.37	0.67	0.317	0.813

**Table 7 ijerph-16-04880-t007:** Multigroup invariance analysis in relation to the gender variable.

Models	χ^2^/df	Δχ^2^	Δgl	CFI	TLI	GFI	RMSEA (IC 90%)
Model 1	0.65	−	−	1.00	1.00	0.99	0.005 (0.004–0.006)
Model 2	0.96	12.05	12	1.00	0.98	0.98	0.006 (0.005–0.007)
Model 3	0.90	23	21	1.00	0.97	0.98	0.006 (0.005–0.007)
Model 4	0.98	77.37	18	0.99	0.97	0.98	0.007 (0.006–0.008)

Note: χ^2^ = chi-squared; df = degrees of freedom; CFI = comparative fit index; TLI = Tucker–Lewis index; SRMR = standardized root-mean-square residual; RMSEA = root-mean-square error of approximation.

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
