# Peer review of "Psychometric Characteristics of the Physical Activity Enjoyment Scale in the Context of Physical Activity in Nature"

_ijerph, 2019, doi:10.3390/ijerph16244880_

Round 1
Reviewer 1 Report
I have no further comments.
Author Response
Thank you so much for the comments. We have improved the writing of English by submitting the document to an edition with specialists. Certificate Attachment

Reviewer 2 Report
Regarding the last version of the article, there is an evident improvement, more coherency, and sustainability of the content provided now. Along the text, there are some words connected, like these examples in the tables "Mybodyfeelsgood", "I finditpleasant" or "changes requested bythe experts" in the document. This should be corrected again by reading carefully the total document.
Author Response

(The authors gave the same response as above.)

Reviewer 3 Report
Review:
This study aimed to examine whether the Physical Activity Enjoyment Scale (PACES) is suitable to use for physical activity in nature. The authors found that sex and school year didn’t show tangible differences. The study is non-experimental, cross-sectional, descriptive and exploratory. There are a number of major flaws in the manuscript that require revision.
Overall
The article is difficult to understand as there are many grammatical errors. Proof reading is required From the abstract, introduction, and discussion, it seems as though this study assessed the psychometric properties of the PACES when used to assess activity in nature. However, it is unclear whether section 2.1.3 is describing how Moreno et al adapted the scale to Spanish or whether the authors of this paper have changed the scale items. If the authors have changed the scale items then this paper is not assessing the PACES scale but a different scale? In the abstract (line 20) and in the discussion (line 80) it says the final model had 5 items. However, in the results section (line 214) and in the conclusion (line 310) it says the final model had 6 items. Table 4 also lists 6 items. Please confirm
Abstract
Line 13 – what does the ‘1’ refer to after school Line 13 – kids is colloquial, consider using ‘children’ Overall it not clear from the abstract what the study is about. What is being tested, what the groups are etc.
Introduction
Line 37-38 – this sentence doesn’t make sense, or is missing something. Are you trying to say that spending more time leisure time engaged in physical activity in nature spaces is important because 75% of Europeans live in urban environments? Line 48-51 – it is confusing about why this is a separate paragraph when it is only two sentences. Line 50-51 – refers to factors mentioned in the previous paragraph. Which factors specifically are being referred to? Line 59 – Introduces the acronym PACES. This has not been referred to previously. Perhaps this could be introduced by the full name and abbreviation when it is first mentioned in line 54. Are 9-13 year old children really classified as adolescents? Line 65-72 – discusses self-efficacy. It is not clear what the links are between self-efficacy and the PACES.
Material and Methods
Line 87 – the end of the introduction says that the sample age range is 9-13 year olds but Line 87 and Line 146 says 12-13 year olds? This is quite a big error that should be amended. Line 104 – describes the scale as a polytomic scale. These are traditionally described as Likert scales. Line 122 – the use of ICC’s is good and the authors should be commended on this Line 156 – the abbreviation CFA is first used but it does not include what the abbreviation stands for.
Results
Line 167 – “particularities” is not a commonly used word, perhaps use “uniqueness” instead. Tables 1 and 5 are not referred to in the text when they are discussed. The Table 1 column headings are not in English. Table 1 and Table 2 includes a number of errors that need to be amended. All tables – there are no spaces between words in the tables. Line 215 – introduces the term PACES-OPACT without explaining what OPACT stands for. Table 4 – scale items 3, 8, and 15 have changed from their previous descriptions. Line 239-243 – mentions analysing gender and school year differences. The result for the gender differences is stated but not the school year differences.
Discussion
It is not clear from the discussion how this research advances the field. It seems like the scale is not valid to use, and thus I am unsure how this paper advances the field. Line 297-307 – why would not being exposed to a lot of physical activity in nature in primary school affect females differently to males? Line 291 – there is no need to reiterate the analytic approach Line 295 – teenagers stated, but have used adolescents throughout. Line 303 – it is not clear how training courses would influence these findings. This argument is not particularly compelling
Conclusion
It is still not clear from the conclusion how this paper advances the field, and the usefulness of this measure in the context of physical activity in nature.Author Response
Thank you so much for the comments. We have improved the writing of English by submitting the document to an edition with specialists. Certificate Attachment
This study aimed to examine whether the Physical Activity Enjoyment Scale (PACES) is suitable to use for physical activity in nature. The authors found that sex and school year didn’t show tangible differences. The study is non-experimental, cross-sectional, descriptive and exploratory. There are a number of major flaws in the manuscript that require revision.
REPONSE:
Thank you very much for the clarification, we have reviewed the design as requested.
Overall
The article is difficult to understand as there are many grammatical errors. Proof reading is required From the abstract, introduction, and discussion, it seems as though this study assessed the psychometric properties of the PACES when used to assess activity in nature. However, it is unclear whether section 2.1.3 is describing how Moreno et al adapted the scale to Spanish or whether the authors of this paper have changed the scale items. If the authors have changed the scale items then this paper is not assessing the PACES scale but a different scale? In the abstract (line 20) and in the discussion (line 80) it says the final model had 5 items. However, in the results section (line 214) and in the conclusion (line 310) it says the final model had 6 items. Table 4 also lists 6 items. Please confirm
REPONSE:
At this point we want to clarify that we have not effectively changed the items of the scale, what has been carried out has been an application of that scale to a different area for which it was built, specifically to the field of physical activities in nature . Therefore, we have modified in point 2.1.3 the application of the PACES scale with 6 Items to physical activities in nature. (Line 135).
We confirm that the scale has 6 Items, we apologize for the error in the transcript. We have corrected the data as requested by the reviewer, in the summary and in the discussion. (Line 22 and 342)
Abstract
Line 13 – what does the ‘1’ refer to after school Line 13 – kids is colloquial, consider using ‘children’ Overall it not clear from the abstract what the study is about. What is being tested, what the groups are etc.
REPONSE
As the reviewer advises, we have modified the word “children” by “students” (line 14) so that it is not colloquially interpreted. We have removed the number "1" (line 14) that was entered in error due to the misconfiguration of the template in the word, we apologize. We have improved the narration and understanding of the summary.
Introduction
Line 37-38 – this sentence doesn’t make sense, or is missing something. Are you trying to say that spending more time leisure time engaged in physical activity in nature spaces is important because 75% of Europeans live in urban environments? Line 48-51 – it is confusing about why this is a separate paragraph when it is only two sentences. Line 50-51 – refers to factors mentioned in the previous paragraph. Which factors specifically are being referred to? Line 59 – Introduces the acronym PACES. This has not been referred to previously. Perhaps this could be introduced by the full name and abbreviation when it is first mentioned in line 54. Are 9-13 year old children really classified as adolescents? Line 65-72 – discusses self-efficacy. It is not clear what the links are between self-efficacy and the PACES.
REPONSE:
In response to the reviewer's revisions, we clarify the changes made:
We improve the narration of line 37-38 now modified (Line 43, 51 and 52). We chose to unify the two paragraphs of the previous lines 48 and 51. We explain the factors to which we refer, (Line 62).
We place the acronym PACES correctly on line 73 which is the first time it appears.
We have used the term “school-age youth” and “adolescents” for a better understanding of the text, and following the classification of the World Health Organization and in other research with populations of these ages (https://onlinelibrary.wiley.com/doi/full/10.1038/oby.2008.296)
As for self-efficacy, we wanted to use this concept by the advice of one of the reviewers. That is why we include quotes that relate self-efficacy to the enjoyment of the use and mention of that term.
Material and Methods
Line 87 – the end of the introduction says that the sample age range is 9-13 year olds but Line 87 and Line 146 says 12-13 year olds? This is quite a big error that should be amended. Line 104 – describes the scale as a polytomic scale. These are traditionally described as Likert scales. Line 122 – the use of ICC’s is good and the authors should be commended on this Line 156 – the abbreviation CFA is first used but it does not include what the abbreviation stands for.
REPONSE:
The age range is 9 to 13 years (Line 88 and 192), we have modified it in the manuscript as well as the polytomic scale by Liker scale as advised by the reviewer (Line 140).
We appreciate the correction of the abbreviation of the Confirmatory Factor Analysis (CFA) and the use of the ICC. (Line 202).
Results
Line 167 – “particularities” is not a commonly used word, perhaps use “uniqueness” instead. Tables 1 and 5 are not referred to in the text when they are discussed. The Table 1 column headings are not in English. Table 1 and Table 2 includes a number of errors that need to be amended. All tables – there are no spaces between words in the tables. Line 215 – introduces the term PACES-OPACT without explaining what OPACT stands for. Table 4 – scale items 3, 8, and 15 have changed from their previous descriptions. Line 239-243 – mentions analysing gender and school year differences. The result for the gender differences is stated but not the school year differences.
REPONSE:
We have modified the term "particularities" by "uniqueness" line 213.
All the clarifications requested by the reviewer are modified referring to Tables 1, 2 and 5. (Line 219, 238 and 288)
We clarify the term OPACT (Outdoor Physical Activity). Line 265. We introduce the mention of gender differences as of school year. (Line 300).
Regarding the elements that have been modified in scale 4 (Line 268), we have reviewed the previous versions and detected the fault. The translation of the manuscript went through several different reviewers and one of them translated the terms 3,8,15 differently to the previous reviewer. For this reason, they appeared different. However, the authors had previously perceived this grammatical error and we intended to modify it as soon as possible.
Discussion
It is not clear from the discussion how this research advances the field. It seems like the scale is not valid to use, and thus I am unsure how this paper advances the field. Line 297-307 – why would not being exposed to a lot of physical activity in nature in primary school affect females differently to males? Line 291 – there is no need to reiterate the analytic approach Line 295 – teenagers stated, but have used adolescents throughout. Line 303 – it is not clear how training courses would influence these findings. This argument is not particularly compelling
REPONSE:
Below we will explain the reasons why this instrument advances in the field of existing research.
1.- An instrument is applied in a different context for which it was created, which is the natural environment.
2.- From its application, it is concluded that the PACES in this context cannot be applied as is, but that some items must be eliminated and / or modified. Therefore, we obtain a new specific instrument for this type of practice.
3.- An invariance is carried out, something that had never before been carried out with any of these instruments, therefore, new data is provided.
Finally, sex and course analysis are done and there are no differences, something that in other works gave different results.Regarding the question asked by the reviewer about why not being exposed to a lot of physical activity in nature in primary school would affect women differently from men, we approached it by commenting that in secondary, the content blocks that They refer to physical activities in the natural environment are mandatory. On the other hand, in the primary age the contents are almost non-existent, they are hardly worked and are not mandatory. (Line 369)
The concepts that have been used are teenagers and preteens. However, the PACES scale has always been used with older students. We advanced in the study using this instrument with students 9 years and older.
We improve in the manuscript how training courses influence these findings. Thanks for the comment.
Conclusion
It is still not clear from the conclusion how this paper advances the field, and the usefulness of this measure in the context of physical activity in nature.
REPONSE:
Following the instructions of the reviewer, we improve the conclusion of our manuscript. For this, we better detail the progress of our study in the scientific literature and the usefulness of the instrument (Line 381)

Round 2
Reviewer 3 Report
The authors should be commended on the improvements made to the manuscript, especially the written expression.
A couple of final comments.
It is still not clear from the manuscript how self-efficacy is linked to PACES. The response of 'a reviewer asked for it' is not sufficient and more explanation and justification is needed. Further discussion is needed as to why females would be affected differently to males.
Author Response
REVIEWER 3
Review Report Form
Open Review
(x) I would not like to sign my review report
( ) I would like to sign my review report
English language and style
( ) Extensive editing of English language and style required
( ) Moderate English changes required
(x) English language and style are fine/minor spell check required
( ) I don't feel qualified to judge about the English language and style
|
Yes |
Can be improved |
Must be improved |
Not applicable |
|
|
Does the introduction provide sufficient background and include all relevant references? |
(x) |
( ) |
( ) |
( ) |
|
Is the research design appropriate? |
( ) |
( ) |
(x) |
( ) |
|
Are the methods adequately described? |
( ) |
(x) |
( ) |
( ) |
|
Are the results clearly presented? |
( ) |
(x) |
( ) |
( ) |
|
Are the conclusions supported by the results? |
( ) |
( ) |
(x) |
( ) |
Comments and Suggestions for Authors
The authors should be commended on the improvements made to the manuscript, especially the written expression.
A couple of final comments.
It is still not clear from the manuscript how self-efficacy is linked to PACES. The response of 'a reviewer asked for it' is not sufficient and more explanation and justification is needed. Further discussion is needed as to why females would be affected differently to males.
Submission Date
17 October 2019
Date of this review
19 Nov 2019 01:26:56
REPONSE TO REVIEWER 3
It is still not clear from the manuscript how self-efficacy is linked to PACES.
Following the opinions of the reviewer, we see it appropriate to modify allusions to the concept of Self-efficacy, since in no case did the authors make reference to it in the original manuscript. It is incorporation was due to the fact that a previous reviewer advised us of its possible inclusion by relating PACES to self-efficacy. At this time, we are satisfied with the contributions of the current reviewer and to meet their demands, the authors chose to eliminate in the manuscript the allusion to the concept of self-efficacy and not link it to the PACES scale, to avoid confusion to the reader. We appreciate the contribution.
The response of 'a reviewer asked for it' is not sufficient and more explanation and justification is needed. Further discussion is needed as to why females would be affected differently to males.
Now on the line 311.
There are different studies that show that girls concern is to shoot boys, the scientific literature has previously shown that part of girls' concerns when it comes to physical activity is not the enjoyment of physical activity, but to perceive a better physical appearance along with controlling your weight. This leads them to not pay attention to enjoyment while doing physical activity, either in nature or outside it [51]. Along the same lines, another investigation [52,53], showed how student boys, unlike student girls, obtained better scores referring to levels of enjoyment when doing physical activity. Another reason is the critical moment of the transition from primary to secondary. The girls reduce the time dedicated to the practice of physical activity [54].
References:
[51] Labrado Sánchez, S. (2011). Diferencias de género en los niveles de práctica de actividad física y hábitos saludables en los adolescentes de Castilla-La Mancha. Eficacia de un programa de intervención. (Tesis Doctoral). Universidad de Castilla-La Mancha.
[52] Cash, T. F., Novy, P. L., y Grant, J. R. (2004). Why do women exercise? Factor analisys and further validation of the reasons for the exercise inventory. Perceptual and motor skills, 78, 539-544.
[53] Cocca, A. (2012). Análisis de nivel de actividad física y los factores relacionados con la salud psicofísica en jóvenes granadinos (Tesis Doctoral). Universidad de Granada.
[54] Fernández Lifante, J. (2016). Analisis del autoconcepto físico, grado de disfrute y percepción del éxito en educación física y su relación con el nivel de actividad física habitual en adolescentes. (Tesis Doctoral). Universidad de Murcia.
Please see the attachment, the paper with change
Thanks

This manuscript is a resubmission of an earlier submission. The following is a list of the peer review reports and author responses from that submission.
Round 1
Reviewer 1 Report
The article is the adaptation of the well-known scale of PACES in the outdoor context, regarding schools in the region of Andaluzia. The physical activity and the increase of exercise outside of the so-called normal spaces are growing substantially, thus is required from the investigators to analyse the effect on the population of such behavioural. The authors are bringing a new vision on the use of the scale PACES. The statistic used along the document is well done, also the results obtained show the correctness of the tools. There are no big flaws in the document, some small errors, but didn't affect the easy reading of the results and the correct understanding of the article.
Author Response
CORRECTIONS
INTRODUCTION
REVIEWER 1
The article is the adaptation of the well-known scale of PAES in the outdoor context, regarding schools in the region of Andaluzia. The physical activity and the increase of exercise outside of the so-called normal spaces are growing substantially, thus is required from the investigators to analyse the effect on the population of such behavioural. The authors are bringing a new vision on the use of the scale PAES.
Studies are cited that deal with the improvement of psychological variables, among which is enjoyment (PAES). Work in traditional contexts such as sports or school has produced satisfactory results. In the study, the research is carried out in the context of physical activity in nature.
“Outdoor activities” has been changed for physical activity in nature, i.e. physical education activities carried out in the midst of nature.
REVIEWER 2
-Make the setting (Andalucia, Spain) clear to readers from the start, since PA has not been increasing everywhere. The abstract makes clear that the study was carried out in Andalucía.
-Clarify that the outdoors means in nature. One can be outdoors in the middle of a city and not have the same restorative effects as are mentioned in the beginning of the Introduction.
“Outdoor activities” has been changed for physical activity in nature, i.e. physical education activities carried out in the midst of nature.
-Unsure what you mean in line 39, page 1. This means more leisure time and doing physical activity in natural environments in order to counter city life as 75% of the European population live in urban environments. 6.- Gascon, M.; Zijlema, W.; Vert, C.; White, M. P.; Nieuwenhuijsen, M. J. Outdoor blue spaces, human health and well-being: A systematic review of quantitative studies. International Journal of Hygiene and Environmental Health,2017,220(8), 1207-122.
-There is room to more comprehensively review the literature on the role of PA enjoyment in PA promotion among youth and how it interacts with other important determinants of PA among youth (e.g., self-efficacy).
Other variables have been taken into account but the work is focused on a single variable; enjoyment. This variable has given very positive results when it has been used in sports contexts, both in adolescents and in adults. In our case, in a context of physical activity in nature and non-sporting content, it is very interesting to seek to justify the application of the scale in the group of child-adolescents.
-I wouldn't think you would provide a hypothesis for psychometric testing. Instead, perhaps you would provide theoretical and/or empirical justification for any a priori factors that you're confirming. Bearing in mind the increase in the number of people doing physical activity in nature, both in its competitive as well as recreational, healthy and leisure aspects, we propose the use of PAES in the first stage of adolescence in the context of physical activity in nature.
-I question the statement about the "serious lack of research on PA and the outdoors.". There is less research works of the study variable in the field of physical activity in nature.
-I'm unclear on what is meant by educational connotations. Please clarify.
The treatment of content carried out in nature is more collaborative and participative than competitive, which is the most frequent area for the application of the enjoyment scale (PAES).
-Please clarify why one would think PACES would need to be tested to see how it performs among youth doing outdoor PA. There's nothing specific to the type of PA in the scale, so why would it potentially perform differently? I think more justification is needed on this point. It seems that the sample being from Spain and across multiple age groups contributes to filling gaps in the literature more than the outdoor PA, although that can still be something you mention.
Although the questionnaire has no modifications in its wording, it is important to point out that the context of physical activities in nature is different from that of educational centers. We are also interested in finding out how contents that are non-traditional in the educational center respond when carried out in context of nature. One of the initial objectives of the study was to check the effect of the contents, such as orienteering, walking in the countryside, climbing etc. in the middle of nature.
METHOD:
-I understand that the youth were performing outdoor PA at the time, but the PACES questions are not specific to outdoor PA activities, so I'm unclear on how it's truly getting at outdoor PA.
In relation at this question, we have added more information in the procedure paragraphs and the final of the Instrumentand adaptation to the Outdoor Physical Activity context.
Also, in line 86 of paper there appears, “The final version obtained was analyzed by a group of four experts (Lynn, 1986) in education and outdoor physical activity in order to ensure the appropriate design of the items in terms of what the construct was intended to measure; the original meaning was maintained [18].” So, the instrument was evaluated by experts in this type of activities.
-I think cross-sectional would more succinctly get across the study design. Line 70-71
We don’t exactly understand this recommendation. Nevertheless, we have added “cross-sectional” in design.
-Sample size in Methods is different from that in the abstract.
The abstract has been corrected.
-Line 96, what is meant by "within theoretical dimensions of the scale"?
The PACES questionnaire has two theoretical dimensions: positive and negative enjoyment. The experts had to analyze each item and evaluate (suitability and understandability) whether the item corresponded with positive or negative dimension, on a scale from 1 (totally disagree) to 4 (totally agree). When an item was evaluated with < 2.5 mean score by experts, it was revised. If an item was not classified by at least three of the four experts as within the theoretical dimensions of the scale it was revised again to analyze potential problems before proposing an alternative wording which would cover the theoretical dimension in a more clear and accurate way. The adaptation of the PACES questionnaire should maintain the same dimensional structure as the original version.
-Clarify the order of events. Did the youth answer the questionnaire before the experts made all of the changes to the scale or vise versa? Then when in the order did 65 students complete the questionnaire?
This paragraph has been improved. The 65 students completed the questionnaire after the experts’ corrections.
-I'd like to see justification for why an EFA and CFA were conducted (particuarly why an EFA was needed).
In spite of having been adapted and validated by experts with CCI and the interquartile range, to ensure statistical validation, we thought it appropriate to carry out an EFA of the PACES instrument,. This analysis is important due several items having been changed and the context being very different. To check the distribution of the items in the dimensions, we considered it appropriate to carry out an exploratory analysis. Nevertheless if the reviewer considers that is not necessary, this analysis could be eliminated. Finally, we carried a confirmatory analysis to verify whether the instrument presented adequate values of reliability and validity once applied in the context of the natural environment. The results helped us to obtain the final instrument.
RESULTS:
I'm unclear on why a non-normal distribution of response on an item means it should be removed from the scale? Lines 154-155
In the analysis of items and homogeneity of scale, we followed the analysis procedure established by Carretero-Dios y Pérez (2007) and Nunnally and Bernstein (1995) for this type of instrument. These authors advise that if several of the indicators of the items give values out of the established ranges to evaluate whether to remove that item and to test if the results of Internal Consistency of the scale are better without the one that with it. For this reason, it is proposed to eliminate the item and test whether the instrument improves or not. This procedure has already been carried out in different works such as: Granero-Gallegos, Baena-Extremera, Pérez-Quero, Ortiz-Camacho and Braco-Amador (2014). Spanish validation of the “Intention to partake in leisure-time physical activity”. Retos, 26,40-45.
DISCUSSION:
You point out that PA enjoyment is important among adolescent girls, but what about boys? Couldn't your findings indicate that PA enjoyment of outdoor PA is equally enjoyed among girls and boys.
No differences were found either in the positive enjoyment or in the negative enjoyment. Although we make reference to significant differences in the gender, we intuit that boys and girls enjoy similar values. (278-279-280)

Reviewer 2 Report
The authors, from in southern Spain, present results of a confirmatory factor analysis of the PACES and how it may vary by gender and grade in school, among youth performing outdoor/in nature physical activity (PA). Major strengths of the study are that gender invariance was analyzed, as well as grade and gender differences. The analytic methods seem sound, although more clarity is need around how it truly captures enjoyment of outdoor PA. Below are suggestions and questions for guiding improvements to the manuscript:
Introduction:
-Make the setting (Andalucia, Spain) clear to readers from the start, since PA has not been increasing everywhere.
-Clarify that the outdoors means in nature. One can be outdoors in the middle of a city and not have the same restorative effects as are mentioned in the beginning of the Introduction.
-Unsure what you mean in line 39, page 1.
-There is room to more comprehensively review the literature on the role of PA enjoyment in PA promotion among youth and how it interacts with other important determinants of PA among youth (e.g., self-efficacy).
-I wouldn't think you would provide a hypothesis for psychometric testing. Instead, perhaps you would provide theoretical and/or empirical justification for any a priori factors that you're confirming.
-I question the statement about the "serious lack of research on PA and the outdoors."
-I'm unclear on what is meant by educational connotations. Please clarify.
-Please clarify why one would think PACES would need to be tested to see how it performs among youth doing outdoor PA. There's nothing specific to the type of PA in the scale, so why would it potentially perform differently? I think more justification is needed on this point. It seems that the sample being from Spain and across multiple age groups contributes to filling gaps in the literature more than the outdoor PA, although that can still be something you mention.
Method:
-I understand that the youth were performing outdoor PA at the time, but the PACES questions are not specific to outdoor PA activities, so I'm unclear on how it's truly getting at outdoor PA.
-I think cross-sectional would more succinctly get across the study design. Line 70-71
-Sample size in Methods is different from that in the abstract.
-Line 96, what is meant by "within theoretical dimensions of the scale"?
-Clarify the order of events. Did the youth answer the questionnaire before the experts made all of the changes to the scale or vise versa? Then when in the order did 65 students complete the questionnaire?
-I'd like to see justification for why an EFA and CFA were conducted (particuarly why an EFA was needed).
Results:
-I'm unclear on why a non-normal distribution of response on an item means it should be removed from the scale? Lines 154-155
Discussion:
-You point out that PA enjoyment is important among adolescent girls, but what about boys? Couldn't your findings indicate that PA enjoyment of outdoor PA is equally enjoyed among girls and boys.
Author Response

(The authors gave the same response as above.)

Round 2
Reviewer 2 Report
My responses are in italics.
There is room to more comprehensively review the literature on the role of PA enjoyment in PA promotion among youth and how it interacts with other important determinants of PA among youth (e.g., self-efficacy).
Other variables have been taken into account but the work is focused on a single variable; enjoyment. This variable has given very positive results when it has been used in sports contexts, both in adolescents and in adults. In our case, in a context of physical activity in nature and non-sporting content, it is very interesting to seek to justify the application of the scale in the group of child-adolescents.
-I understand that the work is focused on PA enjoyment and that it is an important variable. In my opinion, a more comprehensive presentation of how PA enjoyment is known to interact with other variables is still warranted in the Introduction.
-I wouldn't think you would provide a hypothesis for psychometric testing. Instead, perhaps you would provide theoretical and/or empirical justification for any a priori factors that you're confirming. Bearing in mind the increase in the number of people doing physical activity in nature, both in its competitive as well as recreational, healthy and leisure aspects, we propose the use of PAES in the first stage of adolescence in the context of physical activity in nature.
-I don't understand this response.
-I question the statement about the "serious lack of research on PA and the outdoors.". There is less research works of the study variable in the field of physical activity in nature.
-There is research on PA and the outdoors and exposure to green or open spaces. It's true there is likely less of it compared to indoor PA. Reviewing the extant literature in this area, even if it's not a lot, seems important.
-I'm unclear on what is meant by educational connotations. Please clarify.
The treatment of content carried out in nature is more collaborative and participative than competitive, which is the most frequent area for the application of the enjoyment scale (PAES).
-Please clarify why one would think PACES would need to be tested to see how it performs among youth doing outdoor PA. There's nothing specific to the type of PA in the scale, so why would it potentially perform differently? I think more justification is needed on this point. It seems that the sample being from Spain and across multiple age groups contributes to filling gaps in the literature more than the outdoor PA, although that can still be something you mention.
Although
the questionnaire has no modifications in its wording, it is important
to point out that the context of physical activities in nature is
different from that of educational centers. We are also interested in
finding out how contents that are non-traditional in the educational
center respond when carried out in context of nature. One of the initial
objectives of the study was to check the effect of the contents, such
as orienteering, walking in the countryside, climbing etc. in the middle
of nature.
-I'm still unclear on this.
METHOD:
-I understand that the youth were performing outdoor PA at the time, but the PACES questions are not specific to outdoor PA activities, so I'm unclear on how it's truly getting at outdoor PA.
In relation at this question, we have added more information in the procedure paragraphs and the final of the Instrumentand adaptation to the Outdoor Physical Activity context.
Also, in line 86 of paper there appears, “The final version obtained was analyzed by a group of four experts (Lynn, 1986) in education and outdoor physical activity in order to ensure the appropriate design of the items in terms of what the construct was intended to measure; the original meaning was maintained [18].” So, the instrument was evaluated by experts in this type of activities.
-I think cross-sectional would more succinctly get across the study design. Line 70-71
We don’t exactly understand this recommendation. Nevertheless, we have added “cross-sectional” in design.
- a cross-sectional study is a 1-time-point study. It is understood that it is non-experimental and descriptive in nature.
-Sample size in Methods is different from that in the abstract.
The abstract has been corrected.
-Line 96, what is meant by "within theoretical dimensions of the scale"?
The PACES questionnaire has two theoretical dimensions: positive and negative enjoyment. The experts had to analyze each item and evaluate (suitability and understandability) whether the item corresponded with positive or negative dimension, on a scale from 1 (totally disagree) to 4 (totally agree). When an item was evaluated with < 2.5 mean score by experts, it was revised. If an item was not classified by at least three of the four experts as within the theoretical dimensions of the scale it was revised again to analyze potential problems before proposing an alternative wording which would cover the theoretical dimension in a more clear and accurate way. The adaptation of the PACES questionnaire should maintain the same dimensional structure as the original version.
-This would need to be clarified in the manuscript text as well.
-Clarify the order of events. Did the youth answer the questionnaire before the experts made all of the changes to the scale or vise versa? Then when in the order did 65 students complete the questionnaire?
This paragraph has been improved. The 65 students completed the questionnaire after the experts’ corrections.
-I'd like to see justification for why an EFA and CFA were conducted (particuarly why an EFA was needed).
In spite of having been adapted and validated by experts with CCI and the interquartile range, to ensure statistical validation, we thought it appropriate to carry out an EFA of the PACES instrument,. This analysis is important due several items having been changed and the context being very different. To check the distribution of the items in the dimensions, we considered it appropriate to carry out an exploratory analysis. Nevertheless if the reviewer considers that is not necessary, this analysis could be eliminated. Finally, we carried a confirmatory analysis to verify whether the instrument presented adequate values of reliability and validity once applied in the context of the natural environment. The results helped us to obtain the final instrument.
- It is my understanding that if you have a theoretical, empirical (from other studies), and expert guidance on how the factor structure should appear, only a CFA would be warranted. Feel free to provide citations that justify otherwise.
RESULTS:
I'm unclear on why a non-normal distribution of response on an item means it should be removed from the scale? Lines 154-155
In the analysis of items and homogeneity of scale, we followed the analysis procedure established by Carretero-Dios y Pérez (2007) and Nunnally and Bernstein (1995) for this type of instrument. These authors advise that if several of the indicators of the items give values out of the established ranges to evaluate whether to remove that item and to test if the results of Internal Consistency of the scale are better without the one that with it. For this reason, it is proposed to eliminate the item and test whether the instrument improves or not. This procedure has already been carried out in different works such as: Granero-Gallegos, Baena-Extremera, Pérez-Quero, Ortiz-Camacho and Braco-Amador (2014). Spanish validation of the “Intention to partake in leisure-time physical activity”. Retos, 26,40-45.
DISCUSSION:
You point out that PA enjoyment is important among adolescent girls, but what about boys? Couldn't your findings indicate that PA enjoyment of outdoor PA is equally enjoyed among girls and boys.
No differences were found either in the positive enjoyment or in the negative enjoyment. Although we make reference to significant differences in the gender, we intuit that boys and girls enjoy similar values. (278-279-280)
-I see that you repeated that there were no differences by gender identified, and I understand that. Are there studies that also find no difference in PA enjoyment by gender? I suggest citing those. The finding that there are no differences by gender in PA enjoyment of outdoor PA supports the promotion of PA enjoyment across genders.
Additional comment:
-It is my understanding that the abbreviation of the scale is PACES, not PAES, for the English translation.